# Polydeoxyribonucleotide and Shock Wave Therapy Sequence Efficacy in Regenerating Immobilized Rabbit Calf Muscles

**DOI:** 10.3390/ijms241612820

**Published:** 2023-08-15

**Authors:** Yoon-Jin Lee, Yong Suk Moon, Dong Rak Kwon, Sung Cheol Cho, Eun Ho Kim

**Affiliations:** 1Department of Biochemistry, College of Medicine, Soonchunhyang University, Cheonan 31538, Republic of Korea; leeyj@sch.ac.kr; 2Department of Anatomy, Catholic University of Daegu School of Medicine, Daegu 42472, Republic of Korea; ysmoon@cu.ac.kr; 3Department of Rehabilitation Medicine, Catholic University of Daegu School of Medicine, Daegu 42472, Republic of Korea; mysungchul1@gmail.com; 4Department of Biomedical Engineering & Radiology, School of Medicine, Daegu Catholic University, Daegu 42472, Republic of Korea; eh140149@cu.ac.kr

**Keywords:** muscle atrophy, polydeoxyribonucleotides, extracorporeal shock wave therapy, muscle regeneration, rabbit models

## Abstract

This study primarily aimed to investigate the combined effects of polydeoxyribonucleotide (PDRN) and extracorporeal shock wave therapy (ESWT) sequences on the regenerative processes in atrophied animal muscles. Thirty male New Zealand rabbits, aged 12 weeks, were divided into five groups: normal saline (Group 1), PDRN (Group 2), ESWT (Group 3), PDRN injection before ESWT (Group 4), and PDRN injection after ESWT (Group 5). After 2 weeks of cast immobilization, the respective treatments were administered to the atrophied calf muscles. Radial ESWT was performed twice weekly. Calf circumference, tibial nerve compound muscle action potential (CMAP), and gastrocnemius (GCM) muscle thickness after 2 weeks of treatment were evaluated. Histological and immunohistochemical staining, as well as Western blot analysis, were conducted 2 weeks post-treatment. Staining intensity and extent were assessed using semi-quantitative scores. Groups 4 and 5 demonstrated significantly greater calf muscle circumference, GCM muscle thickness, tibial nerve CMAP, and GCM muscle fiber cross-sectional area (type I, type II, and total) than the remaining three groups (*p* < 0.05), while they did not differ significantly in these parameters. Groups 2 and 3 showed higher values for all the mentioned parameters than Group 1 (*p* < 0.05). Group 4 had the greatest ratio of vascular endothelial growth factor (VEGF) to platelet endothelial cell adhesion molecule-1 (PECAM-1) in the GCM muscle fibers compared to the other four groups (*p* < 0.05). Western blot analysis revealed significantly higher expression of angiogenesis cytokines in Groups 4 and 5 than in the other groups (*p* < 0.05). The combination of ESWT and PDRN injection demonstrated superior regenerative efficacy for atrophied calf muscle tissue in rabbit models compared to these techniques alone or saline. In particular, administering ESWT after PDRN injection yielded the most favorable outcomes in specific parameters.

## 1. Introduction

Muscle atrophy, characterized by a decrease in muscle volume, can result from various factors such as malnutrition, nerve dysfunction, and a lack of physical activity. A previous study reported a regression in the capillary network, a vital component responsible for delivering nutrients and immune cells to facilitate the healing and regeneration processes of atrophied muscles [1]. Immobilization, which is often necessary during the period of recovery from injuries or orthopedic surgeries, can lead to detrimental effects on muscle health, including a reduced amount of myofibrillar proteins, impaired metabolism, altered vascularization, and increased fat tissue within the muscles [2,3]. Maintaining muscle mass and function during periods of immobilization is crucial for optimal recovery and rehabilitation.

Muscle atrophy, characterized by a decline in muscle volume, can be attributed to various factors such as malnutrition, nerve dysfunction, physical inactivity, and immobilization. A previous study has highlighted the regression of the capillary network, a vital component responsible for facilitating the healing and regeneration processes of atrophied muscles [1]. Immobilization, commonly required during the recovery phase following injuries or orthopedic surgeries, can adversely affect muscle health, including diminished myofibrillar proteins, impaired metabolic enzyme function, altered vascular and neural supply, and fat substitution within the muscles [2,3]. Preserving muscle mass and function during immobilization is crucial for facilitating optimal recovery and rehabilitation outcomes. Polydeoxyribonucleotide (PDRN) is a compound comprising polymer chains of deoxyribonucleotides. It regulates the expression of cytokines, resulting in a decrease in proinflammatory cytokines, which promote tumor necrosis, and an increase in anti-inflammatory cytokines such as interleukin-10 [4,5]. Additionally, PDRN can stimulate the production and release of vascular endothelial growth factor (VEGF) by augmenting the effects of adenosine on A2A receptors [6,7]. This mechanism plays a vital role in promoting angiogenesis and collagen synthesis, which are crucial for tissue healing and regeneration.

Extracorporeal shock wave therapy (ESWT) is a therapeutic modality that utilizes acoustic waves generated outside the body to target specific anatomical regions. Previous studies have provided evidence of the effectiveness of ESWT in promoting revascularization and reducing pain caused by peripheral nerve injuries [8]. It has also been shown to stimulate axon regeneration and relieve pain [9]. ESWT has been widely used to treat various musculoskeletal conditions, including calcific tendinopathy, lateral epicondylitis, spasticity, plantar fasciitis, and trigger finger [10,11,12,13,14]. Additionally, recent studies have demonstrated the potential of ESWT to accelerate regeneration in cases of both acute and chronic skeletal muscle injuries while promoting increased blood circulation [15,16,17,18]. These references further support the clinical effectiveness of ESWT in enhancing tissue repair and blood flow. A systematic review suggested that ESWT is comparable in effectiveness to botulinum toxin for the treatment of poststroke spastic aggravations [13]. Nonetheless, the available evidence regarding the role of ESWT in muscle regeneration following injuries or atrophy remains limited.

A study on a rat model demonstrated the beneficial effects of ESWT on sciatic nerve function and the prevention of denervation atrophy. This was evidenced by improved sciatic functional index scores compared to control groups [19]. Additionally, ESWT has been shown to stimulate the response mechanisms in skeletal muscle tissue following injury in another rat model [15]. The underlying mechanisms behind these effects are believed to involve mechanotransduction and the upregulation of various growth factors, including insulin-like growth factor, fibroblast growth factor, and VEGF.

In our previous study, we aimed to investigate the effects of combining ESWT with PDRN injection in full-thickness rotator cuff tendon tear rabbit models [20]. The results of our study demonstrated that this combined approach yielded superior outcomes in terms of revascularization, cell proliferation, and fast walking time compared to ESWT alone, PDRN injection alone, or normal saline injection. Notably, applying ESWT prior to the PDRN injection further improved the outcomes. However, no studies have been performed thus far to study the effects of different sequences of ESWT plus PDRN injection on atrophied calf muscles in rabbits immobilized in a cast.

The present study primarily aimed to explore the combined effects of different sequences of ESWT and PDRN injection on atrophied calf muscles in rabbit models subjected to cast immobilization. By elucidating the optimal treatment sequence, we aimed to enhance our understanding of the synergistic effects of these interventions and their potential for promoting muscle regeneration. This article complies with the ARRIVE reporting checklist.

## 2. Results

The results revealed significant differences in the imaging, electrophysiology, and clinical measurements between Group 1 (normal saline) and the other groups (*p* < 0.05, Table 1). The mean rates of atrophy in the right medial and lateral gastrocnemius (GCM) muscle thickness, right calf circumference, and compound muscle action potential (CMAP) amplitude of the right tibial nerve were significantly lower in Groups 4 and 5 compared to the others (*p* < 0.05, Table 1). The values for the same parameters were also significantly lower in Groups 2 and 3 than in Group 1 (*p* < 0.05, Table 1).

Immunohistochemical analysis revealed significant differences between Group 1 (normal saline) and the others (*p* < 0.05, Table 2, Figure 1). The mean cross-sectional area (CSA) of type I medial and lateral GCM muscle fibers was significantly greater in Group 4 (1391.9 ± 23.6 μm^2^ and 1441.2 ± 25.9 μm^2^, respectively) and Group 5 (1137.9 ± 21.1 μm^2^ and 1095.1 ± 22.0 μm^2^) than in the other groups (*p* < 0.05, Table 2, Figure 1). Similarly, the mean CSA of type II muscle fibers was significantly greater in Group 4 (2458.0 ± 50.6 μm^2^ and 2460.4 ± 47.3 μm^2^, respectively) and Group 5 (2046.6 ± 33.8 μm^2^ and 1980.9 ± 27.6 μm^2^) than in the other groups (*p* < 0.05, Table 2, Figure 1). The mean CSA of both GCM muscle fibers (type I, type II, and total) was significantly greater in Groups 2 and 3 than in Group 1 (*p* < 0.05, Table 2).

The VEGF and platelet endothelial cell adhesion molecule-1 (PECAM-1) ratios of the two GCM muscle fibers were significantly higher in Group 4 than in the other groups (*p* < 0.05, Table 2, Figure 1). The VEGF ratios were significantly greater in Group 2 (PDRN) (0.27 ± 0.02 and 0.29 ± 0.02, respectively) and Group 3 (ESWT) (0.32 ± 0.02 and 0.35 ± 0.04, respectively) than in Group 1 (normal saline) (0.17 ± 0.07 and 0.15 ± 0.05, respectively) (*p* < 0.05, Table 2, Figure 1). Similarly, the PECAM-1 ratios of the medial and lateral GCM muscle fibers were significantly higher in Group 2 (0.21 ± 0.02 and 0.28 ± 0.02, respectively) and Group 3 (0.28 ± 0.02 and 0.29 ± 0.04, respectively) than in Group 1 (0.08 ± 0.05 and 0.08 ± 0.06, respectively) (*p* < 0.05, Table 2, Figure 1). The VEGF and PECAM-1 ratios between Groups 2 and 3 did not differ significantly (Table 2).

The densities of PECAM-1, PCNA, and VEGF expression were significantly increased in Group 4 (PDRN + ESWT) and Group 5 (ESWT + PDRN) than in Group 1 (normal saline), Group 2 (PDRN), and Group 3 (ESWT) (*p* < 0.05, Table 3, Figure 2). The densities of PECAM-1 were 1.19 ± 0.11, 1.32 ± 0.09, 1.72 ± 0.08, and 1.77 ± 0.09; the densities of PCNA were 1.30 ± 0.08, 1.50 ± 0.12, 1.85 ± 0.13, and 1.86 ± 0.11; and the densities of VEGF were 1.38 ± 0.13, 1.60 ± 0.07, 1.90 ± 0.07, and 1.88 ± 0.12 in Group 2 (PDRN), Group 3 (ESWT), Group 4 (PDRN + ESWT), and Group 5 (ESWT + PDRN), respectively, when Group 1 (normal saline) was considered 1.

## 3. Discussion

In our study, we aimed to investigate the synergistic effects of PDRN injection and ESWT on atrophied calf muscles in a cast-immobilized rabbit model. Our findings revealed that the combination of PDRN injection and ESWT was superior in promoting muscle regeneration to PDRN injection alone or ESWT alone. The atrophic changes in the GCM muscle were significantly reduced in the groups that received the combined treatment compared to the other groups. Notably, the timing of ESWT application in relation to PDRN injection also influenced the outcomes, with improved results observed when ESWT was applied after PDRN injection. From a clinical perspective, ESWT has proven to be an effective therapeutic approach for the indicated musculoskeletal conditions [10,11,12,13,14,15,16,17,18,21,22,23]. In our study, we used radial ESWT to treat the atrophy of muscles in rabbits. The radial type of ESWT offers certain advantages compared to the focused type. It has less surface infiltration and distributes energy more evenly across the treatment area [24]. A recent systematic review has highlighted the advantages of radial ESWT, including a larger coverage area for therapy, a reduced need for precision targeting, and the elimination of the requirement for additional local anesthesia [25]. Considering that our animal model exhibited the overall atrophy of the GCM muscle, we specifically selected radial ESWT for this study instead of the focused type.

No previous studies have directly explored the application of ESWT for the treatment of atrophied muscles. However, one study used ESWT for the management of myofascial pain syndrome (MPS) [26]. In their research, they administered 1500 pulses of ESWT once a week for two weeks, following a similar protocol to our current study. Their results demonstrated significant improvements in pain relief and subjective disability among patients with MPS [26].

The primary purpose of the study was to evaluate the potential of ESWT in promoting rehabilitation from muscle atrophy induced by immobilization. Shock waves, which are sound waves with high positive pressure amplitudes, possess unique characteristics compared to ultrasonic waves with limited bandwidth. These high-pressure sound waves exhibit increased velocity and generate significant energy by modifying their waveforms. Although the process of ESWT is still a subject of debate, previous studies have suggested that it promotes angiogenesis [27] and tenocyte proliferation [28], which contribute to pain reduction and facilitate functional recovery. Zhang et al. proposed that ESWT is involved in lubricin production in tendons and septa, offering a potential mechanism for reducing tissue wear and tear [29]. In addition, a laboratory study revealed that radial ESWT significantly upregulated the expression of muscle-specific genes, including paired box protein 7, neural cell adhesion molecule, and myogenic factor 5, in muscle cells compared to nontreated cells [30].

In our study, we observed significant improvements in various parameters related to muscle regeneration, including calf muscle circumference, GCM muscle thickness, tibial nerve CMAP, and GCM muscle fiber CSA (type I, type II, and total) in the G4-PDRN + ESWT and G5-ESWT + PDRN groups compared to the other groups. These results are comparable to those of previous studies that have reported the enhanced expression of tissue-specific growth factors following short-term postinjury ESWT [8,9,19,27,28].

Experimental studies conducted in animal models involving injured bones, tendons, and toxin-lesioned limb muscles have demonstrated similar responses to ESWT [20,28,31,32,33]. These studies have revealed that ESWT induces the upregulation of factors such as VEGF, VEGF receptor protein, placental growth factor (PGF), PGF receptor, and transforming growth factor β1, thereby promoting angiogenesis and tissue healing [34]. We observed a significantly higher VEGF and platelet endothelial cell adhesion molecule-1 (PECAM-1) ratio in the GCM muscle fibers of the G4-PDRN + ESWT group than in the other four groups. This finding is consistent with the study by Frey et al., who detected the remarkable repair of injured muscle in a locally VEGF-treated rabbit group, which was attributed to reduced connective tissue and increased muscle fiber count [35]. PECAM-1, known for its involvement in angiogenesis, plays an essential role in healing and regenerative processes [35,36].

Our findings suggest that PDRN injection is beneficial in promoting the recovery of muscle atrophy caused by immobilization. Previous research has demonstrated that PDRN can promote cell growth and migration, stimulate the production of extracellular matrix proteins, and reduce inflammation [4,37]. PDRN can also enter cells and provide pyrimidine or purine rings, which serve as substrates for enzymes involved in nucleic acid production [38]. Purine nucleosides, derived from PDRN, can activate various signaling pathways by binding to specific receptors, activating the reproduction of endothelial cells and neuroglia [39], and displaying synergistic effects with other growth factors [40,41,42]. Research has shown that PDRN promotes the expression of VEGF by activating the adenosine A2A receptor [37]. Moreover, recent research has demonstrated that VEGF promotes healing in diabetic rat feet by inducing angiogenesis and collagen synthesis [38]. Therefore, PDRN may enhance the healing process in atrophied muscles that often exhibit poor vascularization [1,2]. According to our results, we believe that ESWT-induced VEGF expression enhances the regenerative effects of PDRN, as evidenced by a significantly higher VEGF and PECAM-1 ratio observed in the G4-PDRN + ESWT group than in other groups.

The precise mechanism underlying the joint effects of ESWT and PDRN is not yet fully understood. However, this mechanism may offer advantages due to the localized property of ESWT-induced cavitation and its potential impact on endothelial cell membranes [43]. Cavitation refers to the formation of bubbles with conflicting pressures, which can exert stress on cell membranes. This “shearing stress” can increase cell membrane permeability and induce gene expression by activating growth factors [44]. Further research is required to reveal the precise mechanisms of the synergistic effects of ESWT and PDRN in facilitating muscle regeneration and tissue healing.

A previous study [45] revealed no improvement after the combined treatment with cisplatin before or after shock wave treatment. The authors proposed that this result may be attributed to the short-lived effect of ESWT, which lasts for approximately 15 s after the last shock wave, and the permeable nature of cell membranes [46]. In contrast, another study indicated that ESWT enhanced the anesthesia of the rat caudal nerve, indicating the possibility of shock-wave-mediated transdermal drug delivery during the ESWT period [47]. The group receiving EMLA combined with ESWT possessed greater anesthetic capabilities compared to the group applying EMLA after ESWT. These findings are consistent with our study, indicating a synergistic regenerative effect. Our results demonstrate superior effects in certain aspects when ESWT is applied after PDRN injection.

However, our study has several limitations. First, the evaluation period was limited to 4 weeks, including baseline, 2 weeks of immobilization, and PDRN injections. Therefore, the long-term effects of PDRN injections, with or without ESWT, on atrophied muscles should be further investigated. Second, we conducted two sessions of ESWT using a radial-type machine with specific parameters, including a frequency of 3 Hz, an energy density of 0.1 mJ/mm^2^, and 1500 pulses per session. The potential differences in the efficacy of ESWT at varying physical characteristics were not evaluated in our study. Future studies should explore more appropriate interventions for atrophied muscles.

## 4. Materials and Methods

### 4.1. Animal Models

Approval for this study was obtained from the Institutional Animal Care and Use Committee (IACUC) of the Catholic University of Daegu School of Medicine (IRB no.: DCIAFCR-210503-01-Y). Thirty male New Zealand white rabbits, aged 12 weeks and weighing a mean of 3.3 kg, were used. The rabbits were individually placed in steel cages under monitored environmental conditions, including a temperature of 23 ± 2 °C and humidity of 45% ± 10%, and tap water and a commercial rabbit diet were provided for nutrition.

Animals were randomly assigned to five groups, with six rabbits in each group. The assignment was conducted using a random grouping program, which assigned numbers from 1 to 30 to the rabbits. The groups were as follows: Group 1 (G1-Control) received a normal saline injection (0.7 mL); Group 2 (G2-PDRN) received an injection of polydeoxyribonucleotide (PDRN) (0.7 mL); Group 3 (G3-ESWT) underwent ESWT; Group 4 (G4-PDRN + ESWT) received an injection of PDRN (0.7 mL) followed by ESWT; and Group 5 (G5-ESWT + PDRN) underwent ESWT followed by an injection of PDRN (0.7 mL) (Figure 3). The physician, who was blinded to randomization and treatment procedures, conducted all outcome measurements.

### 4.2. Immobilization by Cast

The right lower limbs of rabbits were immobilized with a cast for 2 weeks. The immobilization process involved extending the right knees and ankles by using a splint composed of an adhesive elastic bandage and PVC material (Tensoplast^®^; Smith & Nephew Medical, London, UK). This immobilization technique was conducted following the procedure outlined in a prior study [18].

### 4.3. Injection

The injection involved intramuscularly administering Zoletil^®^ 50 (15 mg/kg, Virbac Korea, Seoul, Republic of Korea) and xylazine (5 mg/kg, Rompun^®^; Bayer Co., Seoul, Republic of Korea). Ultrasound guidance was employed using a 5–13-MHz multifrequency linear transducer (Antares; Siemens Healthcare, Erlangen, Germany). Injections of normal saline or PDRN (0.7 mL each) were administered at both sides of the GCM muscle with the aid of ultrasound imaging (Figure 4A,B). For this purpose, commercial PDRN (Rejuvenex Inj., Polydeoxyribonucleotide sodium, 5.625 mg/3 mL, Pharma Research Product, South, Republic of Korea) was applied. Each injection of 0.35 mL was performed from two sides horizontally, guided by a central reference point. The central reference point was determined as the midpoint between the malleoli of both ankles and the midpoint between the femoral epicondyles, with a horizontal line perpendicular to the intersection of the longitudinal line. The injections were administered repeatedly at the same sites 1 week later.

### 4.4. ESWT

ESWT was performed using a radial-type machine (BTL-5000; BTL, Columbia, SC, USA). Pressure pulses were applied to both sides of the GCM muscle. In the case of Group 4, ESWT was administered immediately following the PDRN injection (Figure 2C,D). Each injection site received 750 shocks, resulting in a cumulative total of 1500 shocks (two sites; energy density = 0.1 mJ/mm^2^; frequency = 3 pulses/s). ESWT was performed repeatedly at the same site 1 week later.

### 4.5. Clinical Procedure

The measurements of all study parameters were performed by a physician who had extensive experience in ultrasound (20 years) and electrophysiology (25 years). The assessment was conducted in a blinded manner to minimize bias. Before euthanasia, motor nerve conduction was evaluated to detect the amplitude of the CMAP on the tibial nerve. The active and reference electrodes were placed at the midpoint of the GCM muscle and the ankle, respectively. Electrical stimuli were applied to the tibial nerve in the popliteal fossa, and the highest CMAP was observed after 7–10 repetitions.

Calf measurements were taken using a tape measure with the knee joints flexed at a 90° angle and the ankle in a relaxed position. Real-time B-mode ultrasound was employed to assess the dimensions of the GCM muscle to the deepest layers of fascia (Figure 5). Longitudinal ultrasound images of the GCM muscle were obtained at the injection sites on both surfaces of the muscle.

To evaluate atrophic changes, the CMAP amplitude, GCM thickness, and calf circumference were calculated with the following formula: ([left side − right side]/left side × 100). The results were presented as the difference in the percentages of atrophic changes on both sides.

### 4.6. Tissue Preparation

At 2 weeks after the removal of cast immobilization, all rabbits were euthanized under general anesthesia. All histological examinations were conducted using the blinded sample, and muscle samples were obtained from the right GCM muscle. The lateral and medial GCM muscle fibers were isolated and stored in neutral-buffered formalin for 1 day. Subsequently, the specimens were embedded in paraffin (Paraplast; Oxford, St. Louis, MO, USA) and divided into 5 mm thick transverse sections for further analysis.

### 4.7. Immunohistochemical Analysis

Immunohistochemical analysis was conducted on the muscle sections to identify type I and type II fibers using monoclonal antimyosin antibodies (Sigma-Aldrich, St. Louis, MO, USA). Specific monoclonal antibodies targeting type I (skeletal, slow) and type II (skeletal and fast) fibers were used. Furthermore, the sections were subjected to immunostaining for angiogenic markers using a polyclonal anti-VEGF antibody (A-20; Santa Cruz Biotechnology, Santa cruz, CA, USA) and an anti-PECAM-1 polyclonal antibody (M-20; Santa Cruz Biotechnology, Santa Cruz, CA, USA).

The paraffin-placed sections were rinsed with PBS, and to inhibit endogenous peroxidases, they were treated with 0.3% H_2_O_2_ in PBS for 30 min. Nonspecific protein binding was prevented by placing them in PBS containing 10% normal horse serum (Vector Laboratories, Burlingame, CA, USA) for 30 min. Then, the respective primary antibodies (diluted at 1:200–1:500) were added at room temperature (23 ± 2 °C) for 2 h, followed by three rinses with PBS. Subsequently, the appropriate secondary antibodies (diluted at 1:100) were added. Biotinylated antimouse immunoglobulin G (IgG) (Vector Laboratories, Burlingame, CA, USA) was stored in the muscle sections for 1 h at room temperature. After three rinses with PBS, the avidin–biotin–peroxidase complex (ABC, Vector Laboratories, Burlingame, CA, USA) was retained on the sections for 1 h, followed by three more rinses. The peroxidase reaction was conducted using 0.05 M Tris-HCl (pH 7.6) with 0.01% H_2_O_2_ and 0.05% 3,3′-diaminobenzidine (DAB, Sigma-Aldrich, St. Louis, MO, USA). Hematoxylin was used for counterstaining, and the slides were mounted. The stained slides were examined using an Axiophot Photomicroscope (Carl Zeiss, Oberkochen, Germany) with an Axio-Cam MRc5 attachment (Carl Zeiss, Oberkochen, Germany).

Histological examinations were conducted in a blinded manner using the Axiophot Photomicroscope (Carl Zeiss, Oberkochen, Germany). Five randomly selected fields were captured from each group for analysis. Morphometric analysis was performed using software (AxioVision SE64 Rel 4.8; Carl Zeiss, Oberkochen, Germany) to evaluate the mean CSA of antimyosin-positive type I and II muscle fibers.

Photographs of 20 randomly selected fields were taken from each group and analyzed with the same software to count the ratio of VEGF- and PECAM-1-positive cells or nuclei per 1000 muscle fibers, as well as the total number of muscle fibers per image.

### 4.8. Tissue Western Blot

Calf tissue samples were stored in radioimmunoprecipitation assay buffer (1× PBS, 1% NP-40, 0.5% sodium deoxycholate, 0.1% sodium dodecyl sulfate (SDS), 10 μg/mL of phenylmethanesulfonyl fluoride, and a protease inhibitor cocktail tablet) for 1 min on ice. The homogenates were then centrifuged at 10,000× *g* for 10 min at 4 °C and then kept at −70 °C for subsequent Western blot analysis. The protein concentration in the supernatants was determined using a BCA assay kit (Thermo Scientific Inc., Waltham, MA, USA) following the manufacturer’s instructions.

For Western blot analysis, 40 μg protein samples were divided by SDS-polyacrylamide gel electrophoresis using NuPAGE 4–12% bis-Tris gels (Invitrogen, Waltham, MA, USA) and delivered to polyvinylidene difluoride (PVDF) membranes (GE Healthcare Life Sciences, Amersham, Bucks, Germany). The latter were then stored in a casein blocking buffer (Sigma-Aldrich, St. Louis, MO, USA), and the antibodies were added. After applying washing and PBS tween-20 buffers, the membranes were saturated for 1 h with antimouse IgG (sc-2005; Santa Cruz Biotechnology, Santa Cruz, CA, USA)-HRP-linked species-specific whole antibody (diluted 1:5000). Enhanced chemiluminescence revealed protein bands on membranes (Promega Corp., Madison, WI, USA), and an anti-β-actin antibody (A2228; Sigma-Aldrich, St. Louis, MO, USA) was added for loading control. The primary antibodies were against PCNA (sc-56; Santa Cruz Biotechnology, Santa Cruz, CA, USA), PECAM-1 (sc-376764; Santa Cruz Biotechnology, Santa Cruz, CA, USA), and VEGF (sc-7269; Santa Cruz Biotechnology, Santa Cruz, CA, USA). The relative density of the protein bands was quantitatively analyzed using TINA software (Version 2.10e).

### 4.9. Statistical Analysis

The data were analyzed using IBM Corp. (Armonk, NY, USA)’s SPSS version 22.0 for Windows. A significance level of 0.05 was used, and *p*-values < 0.05 were considered statistically significant. To determine the appropriate sample size, a pilot study was conducted using the total muscle fiber CSA of the medial GCM as the primary endpoint. One rabbit from each group was included in the pilot study, and five randomly selected fields were evaluated in each group. Based on the effect size of 0.49 received from the pilot study, ANOVA showed that 111 fields were required to attain the power of 95%. Because four fields could be obtained from 1 rabbit, 27 rabbits were initially considered. Accounting for a potential dropout rate of 10%, the final sample size was calculated to be thirty rabbits. Descriptive statistics, including means and standard errors, were calculated, and ANOVA was employed to make the comparisons among the four groups. When significant differences were observed, Tukey’s test was performed for post hoc analysis, and 95% confidence intervals were calculated.

## 5. Conclusions

The results distinctly revealed the superiority of this combined intervention over individual treatments, including ESWT alone, PDRN injection alone, or normal saline injection. Notably, the combined treatment exhibited positive effects on various parameters, such as calf circumference, gastrocnemius muscle thickness, tibial nerve CMAP, and gastrocnemius muscle fiber CSA. These improvements are consistent with the upregulation of proliferating cell nuclear antigen, potentially contributing to enhanced tissue repair and growth.

Furthermore, our findings suggest a mechanistic basis for the observed effects, as indicated by the significant upregulation of VEGF and PECAM-1, which are both associated with angiogenesis. Intriguingly, the sequence of treatment administration played a role in enhancing angiogenesis, with the use of ESWT after PDRN injection demonstrating superior effects.

The sequential application of ESWT after PDRN injection appears to be a promising approach to enhance conservative treatments for muscle atrophy in rabbit models. The positive outcomes observed across various parameters, in addition to the underlying mechanisms that involve angiogenesis, underscore the potential of this combined intervention in promoting tissue repair and recovery. These findings provide valuable insights for advancing therapeutic strategies aimed at addressing muscle atrophy effectively.

## Figures and Tables

**Figure 1 ijms-24-12820-f001:**
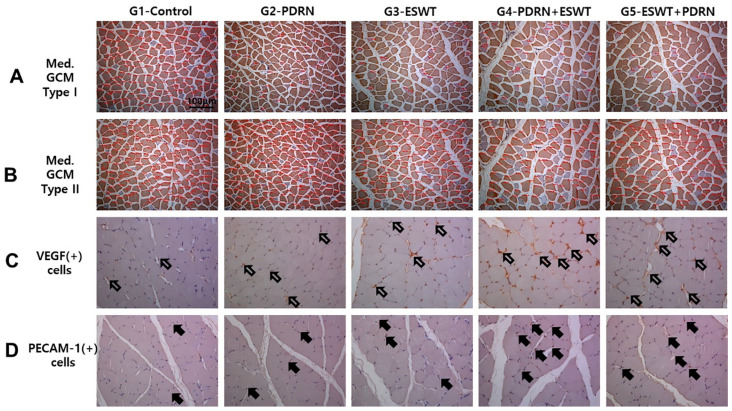
Representative immunohistochemical findings of the immobilized medial GCM muscles stained with monoclonal antimyosin type I (**A**), type II (**B**), anti-VEGF (**C**), and anti-PECAM-1 (**D**) antibodies. The cross-sectional areas (red circles) of the medial head of GCM types I and II muscle fibers were measured using an image morphometry program. The cross-sectional areas of muscle fibers were higher in the G2-PDRN, G3-ESWT, G4-PDRN + ESWT, and G5-ESWT + PDRN groups than in the G1-Control group. VEGF (white arrows) and PECAM-1-positive cells or nuclei (black arrows) and the total number of muscle fibers within each image were counted. The VEGF and PECAM-1 ratios of the medial GCM muscle fibers in the G4-PDRN + ESWT group were significantly higher than those in the other four groups. The scale bar is 50 μm. G1-Control: IC for 2 weeks and 0.7 mL of normal saline injection for 2 weeks after CR; G2-PDRN: IC for 2 weeks and 0.7 mL of PDRN injection for 2 weeks after CR; G3-ESWT: IC for 2 weeks and ESWT after CR; G4-PDRN + ESWT: IC for 2 weeks and 0.7 mL of PDRN injection and ESWT for 2 weeks after CR; G5-ESWT + PDRN: IC for 2 weeks and 0.7 mL of ESWT injection and PDRN for 2 weeks after CR. NS: Normal saline; PDRN: polydeoxyribonucleotide; ESWT: extracorporeal shock wave therapy; GCM: gastrocnemius; VEGF: vascular endothelial growth factor; PECAM-1: platelet endothelial cell adhesion molecule-1; IC: immobilization by cast.

**Figure 2 ijms-24-12820-f002:**
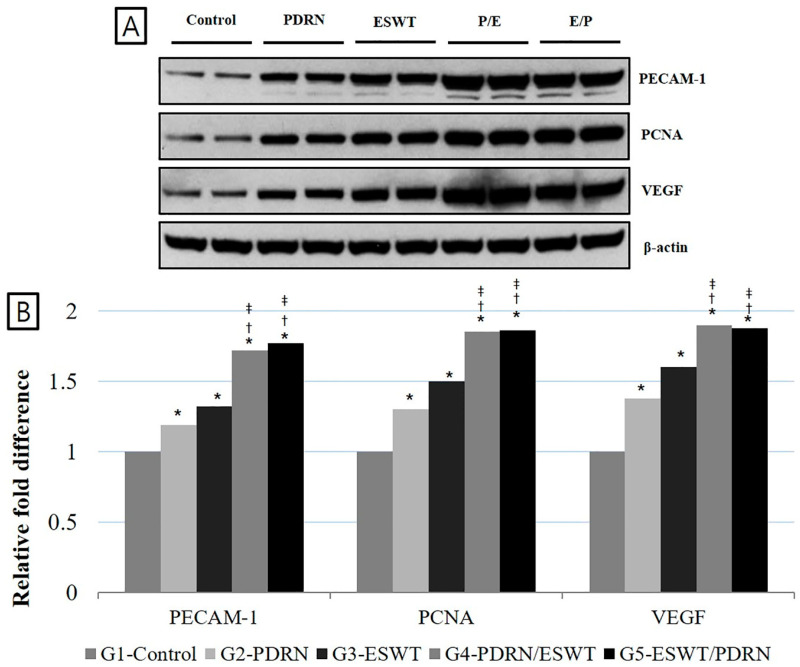
Expression of PECAM-1, PCNA, and VEGF proteins in the medial GCM muscle fibers among the five groups. (**A**) Protein levels of PECAM-1, PCNA, and VEGF. A representative Western blot is shown. (**B**) A relative density (% Control) among the five groups is presented. * *p* < 0.05 one-way ANOVA and Tukey’s post hoc test between Group 1 and Groups 2, 3, 4, and 5. ^†^ *p* < 0.05 one-way ANOVA and Tukey’s post hoc test between Group 2 and Groups 4 and 5. ^‡^ *p* < 0.05 one-way ANOVA and Tukey’s post hoc test between Group 3 and Groups 4 and 5. VEGF: Vascular endothelial growth factor; PECAM-1: platelet endothelial cell adhesion molecule-1; GCM: gastrocnemius; NS: normal saline; PDRN: polydeoxyribonucleotide; ESWT: extracorporeal shock wave therapy; ANOVA: analysis of variance.

**Figure 3 ijms-24-12820-f003:**
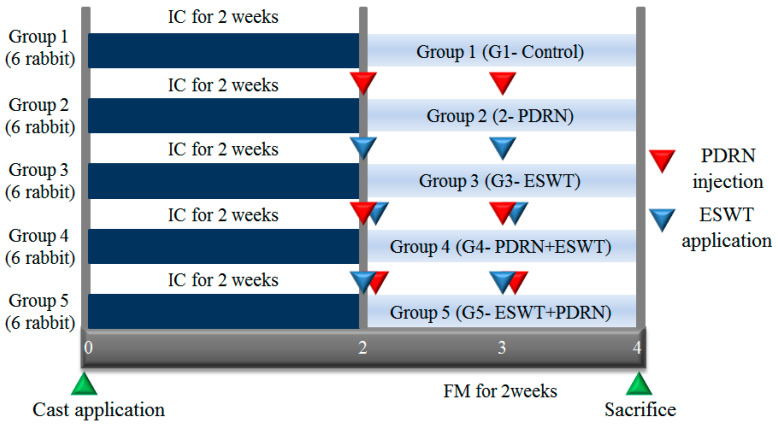
Timeline of the study. A total of 30 rabbits were randomly allocated to five groups. G1-Control: IC for 2 weeks and 0.7 mL of normal saline injection for 2 weeks after CR; G2-PDRN: IC for 2 weeks and 0.7 mL of PDRN injection for 2 weeks after CR; G3-ESWT: IC for 2 weeks and ESWT after CR; G4-PDRN + ESWT: IC for 2 weeks and 0.7 mL of PDRN injection and ESWT for 2 weeks after CR; G5-ESWT + PDRN: IC for 2 weeks and 0.7 mL of ESWT injection and PDRN for 2 weeks after CR. IC: Immobilized by cast; NS: normal saline; PDRN: polydeoxyribonucleotide; ESWT: extracorporeal shock wave therapy; FM: free movement; CR: cast removal. Blue arrowheads: ESWT application; green arrowheads: cast application and sacrifice in sequence; red arrowheads: PDRN injection.

**Figure 4 ijms-24-12820-f004:**
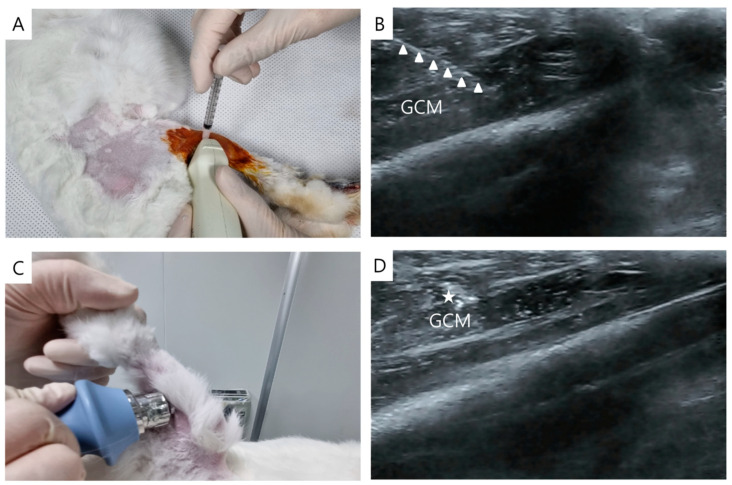
Longitudinal ultrasound image showing PDRN injection at 0.7 mL (**A**) under ultrasound guidance (needle, indicated by arrows) in the right gastrocnemius muscles of a rabbit (**B**). ESWT was then performed under ultrasonic guidance in the region of interest, focusing on the injected PDRN that remained (indicated by an asterisk) near the injection area (**C**,**D**). GCM: Gastrocnemius; PDRN: polydeoxyribonucleotide; ESWT: extracorporeal shock wave therapy.

**Figure 5 ijms-24-12820-f005:**
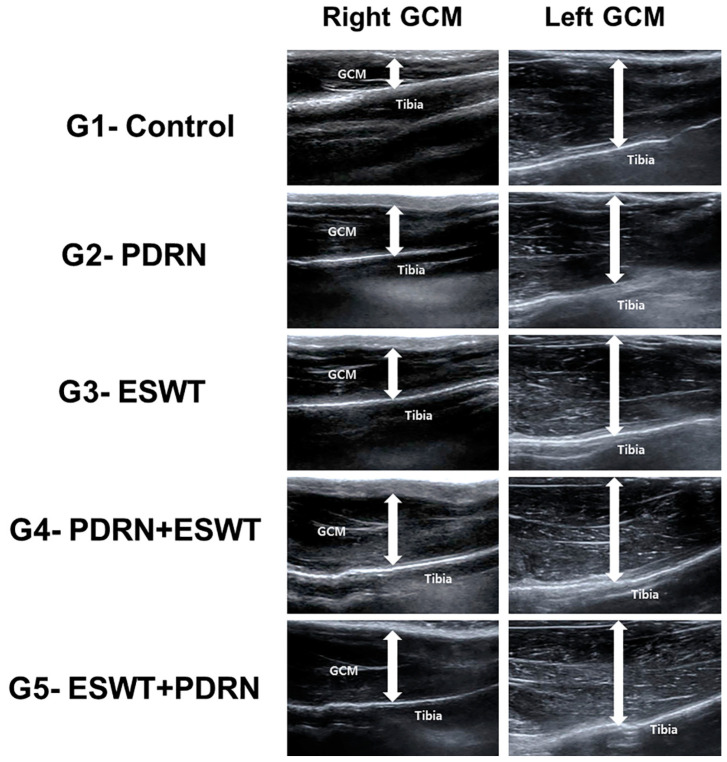
GCM muscle thickness was measured via ultrasound. Thickness was measured as the distance from the superficial aponeurosis to the GCM muscle deep aponeurosis (up–down arrows). Representative longitudinal sonograms of the right and left GCM muscles. The atrophic change in the right GCM muscle was less pronounced in the rabbits in the G4-PDRN + ESWT and G5-ESWT + PDRN groups than in the other three groups. NS: Normal saline; PDRN: polydeoxyribonucleotide; ESWT: extracorporeal shock wave therapy; GCM: gastrocnemius.

**Table 1 ijms-24-12820-t001:** Comparison of the regenerative effects of clinical parameters among the five groups.

Atrophic Changes (%)			
	Circumference of Rt. Calf (cm)	CMAP on Rt. Tibial Nerve (mV)	Rt. GCM Muscle Thickness (mm)
G1-Control	24.1 ± 1.6	25.7 ± 2.0	23.6 ± 1.3
G2-PDRN	21.6 ± 4.0 *	20.2 ± 5.0 *	17.6 ± 3.1 *
G3-ESWT	21.4 ± 3.6 *	20.0 ± 3.9 *	15.4 ± 3.4 *
G4-PDRN + ESWT	5.3 ± 2.7 *^,†,‡^	11.8 ± 2.9 *^,†,‡^	9.3 ± 2.7 *^,†,‡^
G5-ESWT + PDRN	9.8 ± 4.7 *^,†,‡^	13.7 ± 3.3 *^,†,‡^	12.3 ± 3.2 *^,†,‡^

Values are presented as mean ± standard deviation: G1-Control: IC for 2 weeks and 0.2 mL of normal saline injection for 2 weeks after CR; G2-PDRN: IC for 2 weeks and 0.2 mL of PDRN injection for 2 weeks after CR; G3-ESWT: IC for 2 weeks and ESWT after CR; G4-PDRN + ESWT: IC for 2 weeks, 0.2 mL of PDRN injection, and ESWT for 2 weeks after CR; G5-ESWT + PDRN: IC for 2 weeks, ESWT, and 0.2 mL of PDRN injection. IC: Immobilized by the cast; CR: cast removal; PDRN: polydeoxyribonucleotide; ESW: extracorporeal shock wave therapy. * *p* < 0.05 one-way ANOVA and Tukey’s post hoc test between Group 1 and Groups 2, 3, 4, and 5. ^†^
*p* < 0.05 one-way ANOVA and Tukey’s post hoc test between Group 2 and Groups 4 and 5. ^‡^
*p* < 0.05 one-way ANOVA and Tukey’s post hoc test between Group 3 and Groups 4 and 5.

**Table 2 ijms-24-12820-t002:** Comparison of immunohistochemical findings in gastrocnemius muscle fiber among the five groups.

	G1-Control	G2-PDRN	G3-ESWT	G4-PDRN + ESWT	G5-ESWT + PDRN
Medial GCM					
Type I fiber CSA (μm^2^)	287.6 ± 5.5	469.4 ± 12.5 *	716.8 ± 18.9 *	1391.9 ± 23.6 *^,†,‡^	1137.9 ± 21.1 *^,†,‡^
Type II fiber CSA (μm^2^)	443.1 ± 4.5	1554.4 ± 21.4 *	1735.3 ± 26.2 *	2458.0 ± 50.6 *^,†,‡^	2046.6 ± 33.8 *^,†,‡^
Total muscle fiber CSA (μm^2^)	399.0 ± 20.5	1394.0 ± 22.6 *	1542.2 ± 25.6 *	2162.0 ± 42.0 *^,†,‡^	1910.2 ± 31.5 *^,†,‡^
VEGF ratio	0.15 ± 0.07	0.27 ± 0.02 *	0.32 ± 0.02 *	0.59 ± 0.03 *^,†,‡,§^	0.44 ± 0.03 *^,†,‡^
PECAM-1 ratio	0.08 ± 0.05	0.21 ± 0.02 *	0.28 ± 0.02 *	0.59 ± 0.04 *^,†,‡,§^	0.42 ± 0.10 *^,†,‡^
Lateral GCM					
Type I fiber CSA (μm^2^)	286.2 ± 12.3	446.8 ± 15.3 *	697.9 ± 16.2 *	1441.2 ± 25.9 *^,†,‡^	1095.1 ± 22.0 *^,†,‡^
Type II fiber CSA (μm^2^)	420.1 ± 4.9	1450.6 ± 20.3 *	1607.1 ± 17.6 *	2460.4 ± 47.3 *^,†,‡^	1980.9 ± 27.6 *^,†,‡^
Total muscle fiber CSA (μm^2^)	400.73 ± 4.8	1366.6 ± 20.7 *	1481.2 ± 18.5 *	2173.5 ± 40.0 *^,†,‡^	1858.5 ± 26.4 *^,†,‡^
VEGF ratio	0.15 ± 0.05	0.29 ± 0.02 *	0.35 ± 0.04 *	0.57 ± 0.02 *^,†,‡,§^	0.47 ± 0.02 *^,†,‡^
PECAM-1 ratio	0.08 ± 0.06	0.28 ± 0.02 *	0.29 ± 0.04 *	0.58 ± 0.02 *^,†,‡,§^	0.44 ± 0.03 *^,†,‡^

Values are presented as mean ± standard deviation: G1-Control: IC for 2 weeks and 0.2 mL of normal saline injection for 2 weeks after CR; G2-PDRN: IC for 2 weeks and 0.2 mL of PDRN injection for 2 weeks after CR; G3-ESWT: IC for 2 weeks and ESWT after CR; G4-PDRN + ESWT: IC for 2 weeks, 0.2 mL of PDRN injection, and ESWT for 2 weeks after CR; G5-ESWT + PDRN: IC for 2 weeks, ESWT, and 0.2 mL of PDRN injection. IC: Immobilized by cast; CR: cast removal; PDRN: polydeoxyribonucleotide; ESWT: extracorporeal shock wave therapy. * *p* < 0.05 one-way ANOVA and Tukey’s post hoc test between Group 1 and Groups 2, 3, 4, and 5. ^†^
*p* < 0.05 one-way ANOVA and Tukey’s post hoc test between Group 2 and Groups 4 and 5. ^‡^
*p* < 0.05 one-way ANOVA and Tukey’s post hoc test between Group 3 and Groups 4 and 5. ^§^ *p* < 0.05 one-way ANOVA and Tukey’s post hoc test between Group 4 and Groups 1, 2, 3, and 5.

**Table 3 ijms-24-12820-t003:** Comparison of Western blot findings in gastrocnemius muscle fiber among the five groups.

	G1-Control	G2-PDRN	G3-ESWT	G4-PDRN + ESWT	G5-ESWT + PDRN
Relative fold differences					
PECAM-1	1	1.19 ± 0.11 *	1.32 ± 0.09 *	1.72 ± 0.08 *^,†,‡^	1.77 ± 0.09 *^,†,‡^
PCNA	1	1.30 ± 0.08 *	1.50 ± 0.12 *	1.85 ± 0.13 *^,†,‡^	1.86 ± 0.11 *^,†,‡^
VEGF	1	1.38 ± 0.13 *	1.60 ± 0.07 *	1.90 ± 0.07 *^,†,‡^	1.88 ± 0.12 *^,†,‡^

Values are presented as mean ± standard deviation. G1-Control: IC for 2 weeks and 0.2 mL of normal saline injection for 2 weeks after CR; G2-PDRN: IC for 2 weeks and 0.2 mL of PDRN injection for 2 weeks after CR; G3-ESWT: IC for 2 weeks and ESWT after CR; G4-PDRN + ESWT: IC for 2 weeks, 0.2 mL of PDRN injection, and ESWT for 2 weeks after CR; G5-ESWT + PDRN: IC for 2 weeks, ESWT, and 0.2 mL of PDRN injection. VEGF: Vascular endothelial growth factor; PECAM-1: platelet endothelial cell adhesion molecule-1; GCM: gastrocnemius; NS: normal saline; PDRN: polydeoxyribonucleotide; ESWT: extracorporeal shock wave therapy; ANOVA: analysis of variance. * *p* < 0.05 one-way ANOVA and Tukey’s post hoc test between Group 1 and Groups 2, 3, 4, and 5. ^†^
*p* < 0.05 one-way ANOVA and Tukey’s post hoc test between Group 2 and Groups 4 and 5. ^‡^
*p* < 0.05 one-way ANOVA and Tukey’s post hoc test between Group 3 and Groups 4 and 5.

## Data Availability

All data generated or analyzed during this study are included in this published article.

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
