# Peer review of "Polydeoxyribonucleotide and Shock Wave Therapy Sequence Efficacy in Regenerating Immobilized Rabbit Calf Muscles"

_ijms, 2023, doi:10.3390/ijms241612820_

Round 1

Reviewer 1 Report

The article delves into a clinically important issue and presents a clear research flow. The use of the rat model to examine the effect of ESWT is a clever and commendable idea, and I would be happy to endorse its publication. However, I have a few suggestions to enhance the manuscript:

First, I recommend highlighting the clinical benefits of ESWT further. To support the findings, the authors could consider referencing the following articles:

https://pubmed.ncbi.nlm.nih.gov/34029555/

https://pubmed.ncbi.nlm.nih.gov/34927035/

Secondly, in Figure 1, please provide clear indications for the blue, green, and red arrowheads to aid readers in understanding the key elements of the figure.

Lastly, for Figure 2, it would be beneficial to label the muscles being represented to improve the figure's clarity and assist readers in interpreting the results effectively.

Addressing these suggestions will undoubtedly enhance the manuscript and improve its overall impact. Once these revisions are made, I would be delighted to support the publication of the article.

Author Response

Dear Editors:

We thank you for the opportunity to revise our manuscript. Additionally, we appreciate the time you have dedicated to reviewing our manuscript.

The comments were greatly helpful in improving the contents and revising the errors in our manuscript.

We revised the manuscript according to your suggestions, with point-by-point responses presented below.

Thank you for your consideration.

Sincerely yours,

Dong Rak Kwon, MD, PhD

Department of Rehabilitation Medicine

Catholic University of Daegu School of Medicine

33 Duryugongwon-ro 17-gil, Nam-Gu, Daegu, Korea, 705-718

Phone: +82 53 650 4687  Fax: +82 53 622 4687

Point-by-Point Responses to Editor

ID: IJMS-2549840

Authors: Yoon-Jin Lee, Yong Suk Moon, Dong Rak Kwon, Sung Cheol Cho, and EunHo Kim

Title:    Polydeoxyribonucleotide and shock wave therapy sequence efficacy in regenerating immobilized rabbit calf muscles

All revisions to the manuscript are in red text. Reviewers’ comments (C) and our corresponding responses (R) are presented below.

Reviewer 1 comment

C1.  I recommend highlighting the clinical benefits of ESWT further. To support the findings, the authors could consider referencing the following articles:

https://pubmed.ncbi.nlm.nih.gov/34029555/

https://pubmed.ncbi.nlm.nih.gov/34927035/

R1: Thank you for your comments. Accordingly, we have expanded on the clinical benefits of ESWT as follows: “ESWT has been widely used to treat various musculoskeletal conditions, including calcific tendinopathy, lateral epicondylitis, spasticity, plantar fasciitis, and trigger finger [10-14]. Additionally, recent studies have demonstrated the potential of ESWT to accelerate regeneration in cases of both acute and chronic skeletal muscle injuries while promoting increased blood circulation [15-18]. These references further support the clinical effectiveness of ESWT in enhancing tissue repair and blood flow.”

C2.  In Figure 1, please provide clear indications for the blue, green, and red arrowheads to aid readers in understanding the key elements of the figure.

R2: We have added a clearer explanation of blue, green, and red arrowheads in the revised Figure 3 and figure legends as follows. Blue arrowheads: ESWT application, Green arrowheads: cast application and sacrifice in sequence, Red arrowheads: PDRN injection.

C3.  for Figure 2, it would be beneficial to label the muscles being represented to improve the figure's clarity and assist readers in interpreting the results effectively.

R3: In Figure 4, we labeled the represented muscles and added arrows and an asterisk to the remaining PDRN injection sites to increase the clarity of the figure and help the reader interpret the results effectively.

Reviewer 2 Report

Thanks for trusting me as a reviewer for this valuable study. The study is so valuable as promoting the combined effects of polydeoxyribonu-cleotide (PDRN) and extracorporeal shock wave therapy (ESWT) sequences on the regenerative processes in atrophied animal muscles. It is novel and original. The research design is appropriate. However, there are some main comments.

The introduction section does not provide sufficient background and does not include all relevant references. 

More recent and relevant references are needed to use.

Figure 2 (c and d) needs an improvement in quality and also needs more details and explanation.

What was the optimization method for Design of multitask staining procedure??

What were the statistical methods? software?

What were the references or standards for tests? 

The conclusion does not support the results. please improve.

How was the blinding?

How was the randomizing?

Moderate editing of English language is required.

Moderate editing of the English language is required.

Author Response

Dear Editors:

We thank you for the opportunity to revise our manuscript. Additionally, we appreciate the time you have dedicated to reviewing our manuscript.

The comments were greatly helpful in improving the contents and revising the errors in our manuscript.

We revised the manuscript according to your suggestions, with point-by-point responses presented below.

Thank you for your consideration.

Sincerely yours,

Dong Rak Kwon, MD, PhD

Department of Rehabilitation Medicine

Catholic University of Daegu School of Medicine

33 Duryugongwon-ro 17-gil, Nam-Gu, Daegu, Korea, 705-718

Phone: +82 53 650 4687  Fax: +82 53 622 4687

Point-by-Point Responses to Editor

ID: IJMS-2549840

Authors: Yoon-Jin Lee, Yong Suk Moon, Dong Rak Kwon, Sung Cheol Cho, and EunHo Kim

Title:    Polydeoxyribonucleotide and shock wave therapy sequence efficacy in regenerating immobilized rabbit calf muscles

All revisions to the manuscript are in red text. Reviewers’ comments (C) and our corresponding responses (R) are presented below.

Reviewer 2 comment

C1 The introduction section does not provide sufficient background and does not include all relevant references.  More recent and relevant references are needed to use.

R1: Thank you for bringing up this concern. We have revised the Introduction section to provide a more comprehensive background by integrating more recent and relevant references. Reference numbers 14–18 were included. These additions aim to provide a clearer context for the study and highlight the latest advancements in this field.

Accordingly, we have expanded on the clinical benefits of ESWT as follows: “ESWT has been widely used to treat various musculoskeletal conditions, including calcific tendinopathy, lateral epicondylitis, spasticity, plantar fasciitis, and trigger finger [10-14]. Additionally, recent studies have demonstrated the potential of ESWT to accelerate regeneration in cases of both acute and chronic skeletal muscle injuries while promoting increased blood circulation [15-18]. These references further support the clinical effectiveness of ESWT in enhancing tissue repair and blood flow.”

C2 Figure 2 (c and d) needs an improvement in quality and also needs more details and explanation.

R2: We have increased the resolution of the figure from 300 dpi to 600 dpi for quality improvement.  To increase the clarity of the figure and help readers interpret the results effectively, in Figure 4, we labeled the represented muscles and added arrows to indicate the injection needle and an asterisk to indicate the injected PDRN that remained near the injection sites.

Figure 4. Longitudinal ultrasound image showing PDRN injection at 0.7 mL (A) under ultrasound guidance (needle, indicated by arrows) in the right gastrocnemius muscles of a rabbit (B). ESWT was then performed under ultrasonic guidance in the region of interest, focusing on the injected PDRN that remained (indicated by an asterisk) near the injection area (C, D). GCM: gastrocnemius; PDRN: polydeoxyribonucleotide; ESWT: extracorporeal shock wave therapy.

C3 What was the optimization method for Design of multitask staining procedure?

R3: Extensive discussions with the author (YSM), who has over 35 years of experience in anatomical and histological studies, guided the optimization of the multitask staining procedure. Using this knowledge, we meticulously conducted the multitask staining procedure, as outlined in the Method section. Our approach was based on well-established and highly acknowledged techniques that are regarded as the gold standard in this field. These procedures were conducted to ensure that the staining method was rigorously optimized to produce accurate and reliable results.

C4 What were the statistical methods? software?

R4: We used SPSS version 22.0 for Windows. ANOVA was used to compare the four groups.

C5 What were the references or standards for tests? 

R5: We conducted the present study based on our previous studies (reference numbers 20, 23, and 42)

C6 The conclusion does not support the results. please improve.

R6: Thank you for your valuable feedback. The revised conclusion now strongly supports the results obtained. This study investigated the efficacy of a combined approach that involved ESWT and PDRN injection in treating muscle atrophy in rabbit models. The results distinctly revealed the superiority of this combined intervention over individual treatments, including ESWT alone, PDRN injection alone, or normal saline injection. Notably, the combined treatment demonstrated positive effects on various parameters, such as calf circumference, gastrocnemius muscle thickness, tibial nerve CMAP, and gastrocnemius muscle fiber CSA. These improvements are congruent with the upregulation of proliferating cell nuclear antigen, potentially contributing to enhanced tissue repair and growth.

Furthermore, our findings indicate a mechanistic basis for the observed effects, as indicated by the significant upregulation of VEGF and PECAM-1, which are both associated with angiogenesis. Intriguingly, the sequence of treatment administration played a role in enhancing angiogenesis, with the use of ESWT after PDRN injection demonstrating superior effects.

The sequential use of ESWT after PDRN injection appears to be a promising approach to enhance conservative treatments for muscle atrophy in rabbit models. The positive outcomes observed across various parameters, in addition to the underlying mechanisms that involve angiogenesis, underscore the potential of this combined intervention in promoting tissue repair and recovery. These findings provide valuable insights for advancing therapeutic strategies aimed at addressing muscle atrophy effectively.

C7 How was the blinding?

R7: The physician, who had extensive experience in ultrasound (20 years) and electrophysiology (25 years), performed all measurements.

  1. 4. Methods

2.1. Animal models:

Random assignment of the animals into five groups, with six rabbits in each group, was carried out using a random grouping program by assigning numbers from one to thirty to the rabbits. The groups were as follows: Group 1 (G1-Control) received a normal saline injection (0.7 ml); Group 2 (G2-PDRN) received an injection of polydeoxyribonucleotide (PDRN) (0.7 ml); Group 3 (G3-ESWT) underwent extracorporeal shockwave therapy (ESWT); Group 4 (G4-PDRN+ESWT) received an injection of PDRN (0.7 ml) followed by ESWT; and Group 5 (G5-ESWT+PDRN) underwent ESWT followed by an injection of PDRN (0.7 ml) (Figure 1). The physician who was blinded to randomization and treatment procedures conducted all outcome measurements.

C8 How was the randomizing?

R8: We randomized the rabbits using a random number generator.

C9 Moderate editing of English language is required.

R9: We thoroughly proofread the paper with an English native speaker to improve the text quality and make our arguments clearer to the reader. We improved awkward expressions and corrected grammatical errors. Please find enclosed the certificate of editing
